# FlowMM: Cross-Modal Information Flow Guided KV Cache Merging for Efficient Multimodal Context Inference

## Abstract

Traditional KV cache eviction strategies, which discard less critical KV-pairs based on attention scores, often degrade generation quality, causing context loss or hallucinations. Recent efforts shift toward KV merging, merging eviction tokens with retention tokens based on similarity. However, in multimodal scenarios, distributional biases across modality tokens and attentional biases in cross-modal interactions limit its effectiveness. This work introduces FlowMM, an adaptive framework for cross-modal information flow-guided multimodal KV cache merging. FlowMM leverages cross-modal information flow to dynamically apply layer-specific merging strategies, capturing modality-specific patterns while preserving contextual integrity. Furthermore, we introduce a sensitivity-adaptive token matching mechanism that jointly evaluates token similarity and task-critical sensitivity, merging low-risk tokens while safeguarding high-sensitivity ones. Extensive experiments across diverse leading MLLMs show that FlowMM reduces KV cache memory by 80% to 95% and decoding latency by 1.3-1.8×, while maintaining competitive task performance.

## 1 Introduction

Multimodal large language models (MLLMs) based on transformer architecture (Wang et al., 2024a; Liu et al., 2023; Chen et al., 2024b; OpenAI, 2024) have revolutionized the integration of visual and textual understanding, enabling sophisticated cross-modal reasoning across tasks such as visual question answering, image captioning, and multimodal dialogue. Unlike traditional language models that process sequential text tokens, MLLMs face unique computational challenges due to their heterogeneous input modalities. Visual inputs, typically encoded as high-dimensional feature maps or patch embeddings (Dosovitskiy et al., 2021; Liu et al., 2023), generate substantially longer token sequences than their textual counterparts. This multimodal architecture significantly amplifies the memory burden of key-value (KV) cache. Addressing these multimodal-specific memory efficiency challenges has become paramount for scaling MLLMs to real-world applications with limited computational resources. A promising approach involves selectively retaining only the most critical tokens while evicting others (Zhang et al., 2023a; Li et al., 2024; Xiao et al., 2023b). Though effective for memory compression, such eviction-based approaches rely heavily on current token importance assessments. This risks unintentionally and permanently discarding tokens essential for subsequent decoding steps, leading to contextual degradation.

Recently, KV cache merging techniques (Zhang et al., 2024; Wang et al., 2024b) have gained attention as an alternative strategy. By consolidating eviction-targeted states into compact representations, these methods preserve richer contextual information. However, existing merging solutions primarily target unimodal LLMs and exhibit suboptimal performance when naively applied to multimodal scenarios. Specifically, multimodal tokens exhibit significant distributional divergence (Li et al., 2023), and indiscriminate merging risks information confusion or semantic distortion. Furthermore, intricate cross-modal interactions within MLLMs (Alayrac et al., 2022) necessitate careful consideration of attention patterns and dependencies during merging. These challenges render traditional unimodal approaches inadequately equipped to accurately identify mergeable state sets without critical information loss.

To address these challenges, we investigate the impact of KV cache merging in multimodal settings. We empirically find that merging performance varies dramatically across different model layers, revealing significant differences in how layers process heterogeneous modalities. This observation aligns with established principles of attention information flow in prior works (Zhang et al., 2025; Ye et al., 2025). Building on this insight, we introduce FlowMM, an adaptive framework for cross-modal information flow-guided multimodal KV cache merging. FlowMM proactively captures cross-modal interaction patterns across transformer layers by analyzing multimodal attention flow, then dynamically applies layer-specific merging strategies to consolidate critical contextual information.

Further, we identify highly sensitive tokens whose merging substantially degrades model performance. We posit these tokens carry task-critical information vulnerable to corruption during merging. To mitigate this, FlowMM incorporates a sensitivity-adaptive token matching strategy that jointly evaluates token similarity and sensitivity, prioritizing low-sensitivity tokens for merging while preserving high-sensitivity, information-rich tokens. Notably, FlowMM operates without fine-tuning and serves as a plug-and-play solution, delivering adaptive KV cache compression optimized for multimodal contexts.

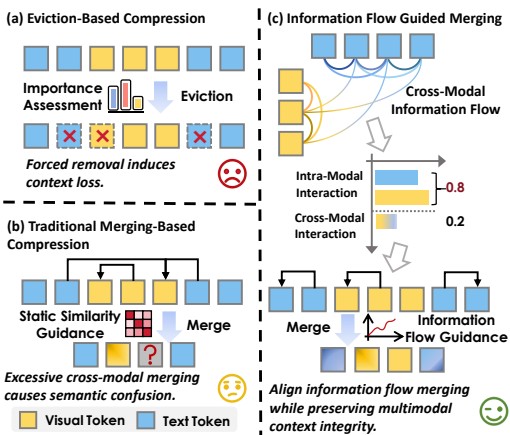

Figure 1: Comparison between Eviction-Based Compression (a), Traditional Merging-Based Compression (b), and our Cross-Modal Information Flow Guided Merging (c).

We conduct extensive experiments with leading MLLMs, including Qwen2.5-VL (Bai et al., 2025), InternVL-2.5 (Chen et al., 2024b), and MobileVLM-V2 (Chu et al., 2024). Their performance is evaluated on MileBench (Song et al., 2024), a comprehensive benchmark encompassing diverse multimodal long-context tasks: temporal multi-image reasoning, semantic multi-image understanding, needle-in-a-haystack retrieval, and complex image search. FlowMM consistently outperforms strong baselines at equivalent KV cache sparsity levels. Specifically, FlowMM achieves a 1.3x to 1.8x reduction in decoding latency while simultaneously reducing the KV cache memory footprint by 80% to 95%. Crucially, these significant efficiency gains are attained while maintaining competitive performance across all evaluated multimodal context tasks.

Overall, our contributions can be summarized as follows:

- We introduce FlowMM, an adaptive framework for cross-modal information flow-guided multimodal KV cache merging. FlowMM dynamically analyzes cross-modal attention flow patterns across layers and employs layer-specific merging strategies, effectively consolidating critical multimodal context.

- To prevent corrupting task-critical information, FlowMM incorporates a sensitivity-adaptive token matching strategy. This jointly evaluates token similarity and sensitivity, merging low-sensitivity tokens while preserving high-sensitivity, information-rich ones.

- We validate FlowMM through extensive experiments across various multimodal context tasks. The results demonstrate that FlowMM reduces KV cache memory usage by up to 80% while consistently surpassing the performance of existing compression methods.

## 2  RELATED WORK

### 2.1  EFFICIENT INFERENCE FOR LARGE LANGUAGE MODELS

Achieving efficient inference in large-scale models requires optimizing three critical resources: model parameters, activation memory, and KV cache size. For parameter reduction, post-training quantization techniques including GPTQ (Frantar et al., 2022), AWQ (Lin et al., 2024), and SmoothQuant (Xiao et al., 2023a) significantly compress weight bitwidth with minimal accuracy

loss, while pruning methods such as SparseGPT (Frantar & Alistarh, 2023) and Wanda (Sun et al., 2023) remove redundant weights or channels. Activation optimizations similarly employ quantization exemplified by ZeroQuant (Yao et al., 2022) alongside dynamic sparsity strategies.

However, the memory footprint of KV cache during autoregressive decoding escalates dramatically with sequence length and model scale, emerging as a dominant bottleneck. This challenge intensifies in MLLMs, where vision encoders generate extensive visual tokens, significantly exacerbating KV cache pressure in subsequent LLM decoding stages.

To alleviate MLLM input burdens, prominent strategies focus on reducing visual tokens fed to the LLM. MobileVLM (Chu et al., 2023) employs aggressive compression via lightweight projections and pooling; LLaVA-PruMerge (Shang et al., 2024) and MADTP (Cao et al., 2024) introduce adaptive token pruning/merging mechanisms ; FastV (Chen et al., 2024a) combines early-layer attention with late-layer pruning. These approaches effectively shorten input sequences, indirectly mitigating downstream KV cache demands. Critically, existing methods primarily target visual token reduction before LLM processing and often require task-specific fine-tuning. They address KV cache efficiency only as a secondary effect, lacking direct optimization mechanisms. Specialized techniques for compressing KV caches in MLLMs remain an underexplored research frontier.

## 2.2 KV Cache Compression

To address the critical bottleneck of KV cache memory overhead in MLLM inference, we systematically examine three dominant compression paradigms: eviction, quantization, and merging.

Eviction methods aggressively prune KV states by retaining only salient tokens while discarding others. Representative approaches like H2O (Zhang et al., 2023a) and SnapKV (Li et al., 2024) leverage attention scores to prioritize token retention. However, this irreversible information loss frequently induces context fragmentation and hallucinations, severely compromising long-context modeling capabilities. Quantization techniques preserve full context coverage while reducing bit precision. MiKV (Yang et al., 2024) retains low-precision representations of evicted states, while KIVI (Liu et al., 2024b) and GEAR (Kang et al., 2024) employ channel-wise key and token-wise value quantization. Although memory-efficient, these methods typically require specialized retraining or calibration, hindering seamless integration with existing LLM infrastructures.

Merging strategies condense multiple KV states into compact representations, minimizing performance degradation under memory constraints. MiniCache (Liu et al., 2024a) exploits inter-layer similarity for intra-layer compression, and CaM (Zhang et al., 2024) aggregates eviction candidates into preserved states. Crucially, these single-modal optimizations exhibit limited efficacy in MLLMs due to cross-modal distribution shifts and attention pattern divergence, failing to preserve modality-specific information fidelity.

## 3 METHOD

### 3.1 PRELIMINARY

MLLMs follow an autoregressive inference paradigm similar to text-only LLMs during the reasoning process, but they need to process heterogeneous input sequences containing both textual and visual tokens. Considering a multimodal input prompt consisting of interleaved text and image tokens, we can represent the concatenated prompt embeddings as :

$$\mathbf{X} = \{\mathbf{X}_1^{\mathrm{T}}, \mathbf{X}_1^{\mathrm{I}}, \ldots, \mathbf{X}_N^{\mathrm{T}}, \mathbf{X}_M^{\mathrm{I}}\} \in \mathbb{R}^{L_{\mathrm{P}} \times d} \tag{1}$$

where $\mathbf{X}^{\mathrm{T}}$ and $\mathbf{X}^{\mathrm{I}}$ denote textual and visual embeddings respectively, $L_{\mathrm{P}}$ is the total prompt length, and $d$ is the hidden dimension. In the prompt encoding phase, the key and value tensors for each transformer layer are computed as:

$$\mathbf{K}_0 = \mathbf{X}\mathbf{W}^K, \quad \mathbf{V}_0 = \mathbf{X}\mathbf{W}^V \tag{2}$$

where $\mathbf{W}^K, \mathbf{W}^V \in \mathbb{R}^{d \times d}$ are the projection matrices. In the generation phase, the model sequentially produces tokens while maintaining and updating the KV cache. At generation step $t$, the KV cache is updated by concatenating new key-value pairs:

$$\mathbf{K}_t = [\mathbf{K}_{t-1}, \mathbf{k}_t], \quad \mathbf{V}_t = [\mathbf{V}_{t-1}, \mathbf{v}_t] \tag{3}$$

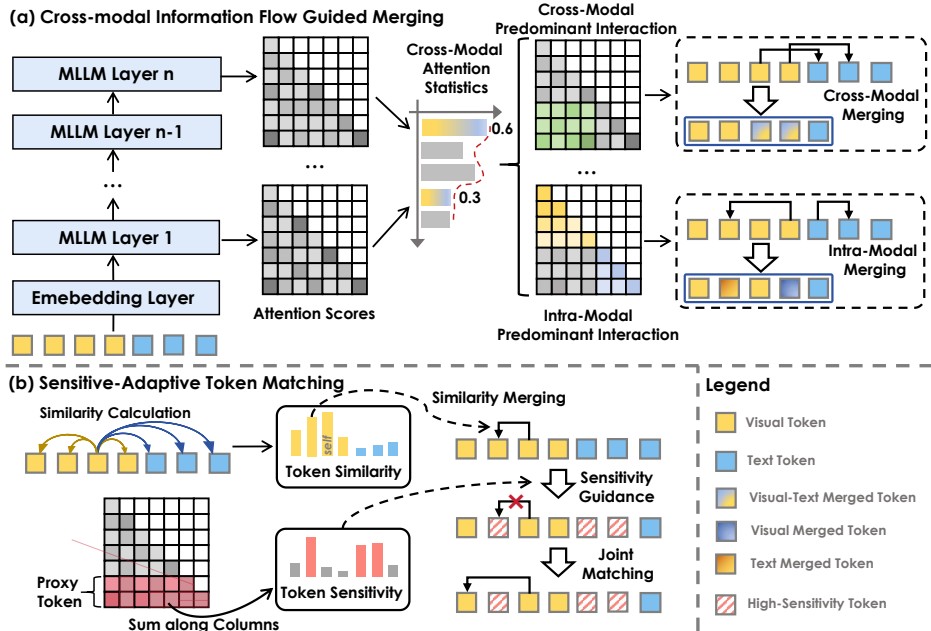

Figure 2: **Overview of FlowMM.** (a) Cross-modal information flow analysis determines whether each layer exhibits predominantly cross-modal or intra-modal interactions, enabling layer-specific merging strategies. (b) Sensitivity-adaptive token matching jointly considers token similarity and sensitivity scores, preserving high-sensitivity tokens while merging similar low-sensitivity tokens to maintain critical contextual information.

where $\mathbf{k}_t = \mathbf{x}_t \mathbf{W}^K$ and $\mathbf{v}_t = \mathbf{x}_t \mathbf{W}^V$ represent the key and value projections of the new token embedding $\mathbf{x}_t$. Finally, the attention output for the current step is computed as:

$$\mathbf{o}_t = \mathrm{Softmax}\left(\frac{\mathbf{q}_t \mathbf{K}_t^\top}{\sqrt{d}}\right)\mathbf{V}_t \tag{4}$$

While KV cache eliminates redundant computations in autoregressive generation, it creates substantial memory overhead as the sequence length grows. This challenge is especially severe in multimodal settings due to long visual token sequences from image encoders. KV cache Merge addresses this by compressing the cache through merging semantically similar key-value pairs, preserving vital attention information. The core principle of KV cache merge involves identifying tokens with high semantic similarity and consolidating their representations. This process can be formulated as:

$$\mathbf{K}^{\mathrm{merged}} = f_{\mathrm{merge}}(\mathbf{K}_t, \mathbf{S}), \quad \mathbf{V}^{\mathrm{merged}} = g_{\mathrm{merge}}(\mathbf{V}_t, \mathbf{S}) \tag{5}$$

where $\mathbf{S} \in \mathbb{R}^{L_t \times L_t}$ represents a similarity matrix that captures pairwise relationships between tokens, and $f_{\mathrm{merge}}$, $g_{\mathrm{merge}}$ are merging functions that aggregate similar representations. This compression strategy effectively reduces memory complexity from $O(L_{\mathrm{p}} + t)$ to $O(L_{\mathrm{compressed}})$ where $L_{\mathrm{compressed}} \ll L_{\mathrm{p}} + t$, enabling efficient processing of long-context multimodal sequences. However, the success of KV cache merge critically depends on maintaining the integrity of multimodal information while achieving significant compression ratios, which presents unique challenges in the context of heterogeneous token representations.

## 3.2 OBSERVATION

In this section, we explore how attention flow patterns influence KV cache merging of MLLMs in multimodal scenarios, presenting experimental findings. The study is conducted on the Qwen2.5-VL-7B (Bai et al., 2025).

### 3.2.1 CROSS-MODAL INFORMATION PATTERNS.

Unlike traditional unimodal LLMs, MLLMs jointly process encoded visual and textual tokens to solve multimodal tasks, where cross-modal interactions generate responses. We first analyze pat-

terns in cross-modal information transfer. Specifically, we conduct zero-shot inference on three datasets: ALFRED, MMCoQA, and Text Needle In A Haystack, measuring the proportion of attention scores allocated to tokens originating from the heterogeneous modality. All attention scores are aggregated through head-wise averaging.

As illustrated in Figure 3(a), our analysis reveals a striking divergence in cross-modal information flow patterns across the layers of MLLMs. This pattern exhibits consistent trends across diverse tasks. In the shallower layers, token interactions are predominantly intra-modal, characterized by significantly lower cross-modal attention scores. Conversely, deeper layers undergo a distinct shift, where intermodal interactions become dominant, corresponding to a substantial increase in cross-modal attention scores. We posit that shallow layers primarily facilitate low-level, unimodal feature extraction, while deeper layers progressively specialize in cross-modal fusion and higher-level semantic abstraction. This inherent functional disparity renders prior KV cache compression methods that apply uniform merging strategies across all layers inherently suboptimal. Consequently, our findings motivate the development of layer-specific merging schemes explicitly designed to align with these distinct cross-modal dynamics.

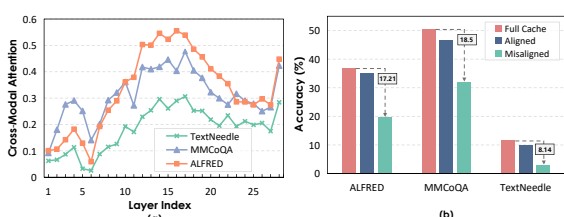

Figure 3: (a) Layer-wise divergence in cross-modal attention. (b) The performance comparison between full cache and information-flow merging.

### 3.2.2 CROSS-MODAL INFORMATION FLOW MERGING.

To investigate the significance of cross-modal information patterns for multimodal KV cache merging, we empirically design merging strategies across diverse tasks. Specifically, we implement aligned information flow merging, performing intra-modal merging in layers with low cross-modal interaction and inter-modal merging in layers with high interaction. We contrast this with misaligned merging (applying intra-modal merging at high-interaction layers and inter-modal merging at low-interaction layers) and compare both against full cache.

As illustrated in Figure 3(b), aligned merging achieves performance comparable to full caching, while misaligned merging causes significant degradation. For instance, in the ALFRED task, misaligned merging only attain approximately 50% of the accuracy of full cache. We posit that reverse information flow merging may cause modal information confusion or semantic distortion. For example, prematurely merging across modalities without sufficient interaction between heterogeneous modality tokens at the shallow layers could disrupt the original modality representation of the tokens. This insight indicates that effective multimodal KV cache merging requires alignment with the inherent cross-modal information flow.

### 3.3 FLOWMM

#### 3.3.1 INFORMATION FLOW GUIDED MERGING.

Cross-modal information flow characterizes the interaction intensity between heterogeneous modality tokens across different layers within MLLMs. Neglecting this flow significantly impairs KV cache merging performance. To address this, we introduce a Multimodal Information Flow-Guided KV Cache Merging strategy. This approach dynamically adjusts layer-specific merging strategy by quantifying cross-modal interaction intensity at each layer. Specifically, we define the cross-modal interaction ratio for a layer as the proportion of attention scores allocated to heterogeneous modality tokens:

$$\rho^l = \frac{1}{H}\sum_{h=1}^{H}\frac{A_{v \to t}^{l,h} + A_{t \to v}^{l,h}}{A^{l,h}}, \tag{6}$$

where $H$ denotes the number of attention heads, and $A^{l,h}$ represents the sum of attention scores for the $h$-th attention head in the $l$-th layer. We define the cross-modal attention scores $A_{v \to t}^{l,h}$ and $A_{t \to v}^{l,h}$ as follows:

$$A_{v \to t}^{l,h} = \sum_{v \in V}\sum_{t \in T}\alpha_{v \to t}^{l,h}, \quad A_{t \to v}^{l,h} = \sum_{t \in T}\sum_{v \in V}\alpha_{t \to v}^{l,h}, \tag{7}$$

where $V$ and $T$ denote the sets of visual tokens and text tokens, respectively, and $\alpha_{i \to j}$ represents the attention score from the $i$-th token to the $j$-th token. Then, we introduce a cross-modal merging threshold $\theta$ to dynamically guide merging strategies. When the cross-modal attention interaction ratio $\rho^l$ at layer $l$ exceeds $\theta$, significant cross-modal interactions exist, warranting cross-modal merging. Conversely, if $\rho^l$ falls below $\theta$, the layer predominantly processes intra-modal information, and we advocate more conservative intra-modal merging.

A crucial step in performing KV cache merging is identifying the sets that will be merged. Directly clustering and merging the KV cache sets is computationally expensive and may fail to leverage task-specific information, potentially leading to the disruption of context that is relevant to the task. Therefore, in this work, we first evaluate token importance. Previous studies have shown that using cumulative attention to evaluate token importance can be biased. To address this, we opt to use proxy tokens to provide a more equitable assessment of token importance:

$$\mathcal{I}^{l,h}(i) = \sum_{j \in \mathcal{P}} \alpha_{j \to i}^{l,h}, \tag{8}$$

where $\mathcal{P}$ denotes the set of proxy tokens. We designate a small subset of tokens near the end of the prompt as proxies, as these tokens typically capture task-specific contextual information. We select the top-$B$ KV pairs with highest token importance to form a pivot set $\mathrm{K}^p$ capturing the most critical task information. The non-pivotal set $\mathrm{K}^n$ are then merged into the pivot set to minimize excessive loss of contextual information.

### 3.3.2 SENSITIVITY-ADAPTIVE TOKEN MATCHING.

Building upon the flow-guided multimodal KV cache merging, we introduce a critical component to address the risk of semantic corruption during state consolidation: Sensitivity-Adaptive Token Matching. This method specifically targets the identification and preservation of highly sensitive tokens crucial for maintaining model performance.

We define the sensitivity of a token within the current context as its contribution to preserving the model's output fidelity. A token is deemed highly sensitive if merging its KV state with others results in a substantial negative impact on the accuracy or relevance of subsequent model generations. Sensitivity is thus intrinsically linked to the token's unique informational value and its role in the multimodal reasoning chain. However, directly measuring the impact of merging each token through repeated perturbation tests during inference, incurs prohibitive computational costs for real-time scenarios. To address this, we propose attention scores as an efficient sensitivity metric. Attention scores directly quantify a token's influence on the current generation step, offering a near-zero-overhead approximation of sensitivity.

We assess the similarity between $\mathrm{K}^p$ and $\mathrm{K}^n$ by employing cosine similarity:

$$u_{i,j} = \frac{k_i^T k_j}{\|k_i\| \, \|k_j\|}, \tag{9}$$

where $u_{i,j}$ represents the cosine similarity between token $i$ and token $j$, and $\|\cdot\|$ is the norm. We then identify the nearest token in $\mathrm{K}^p$ for each token in $\mathrm{K}^n$, as formulated below:

$$k_*^{\text{nearest}} = \underset{\substack{j \in K^p \\ \mathcal{I}_j \leq \tau}}{\text{Argmax}}(u_{i,j}), \tag{10}$$

where $\tau$ denotes the sensitivity threshold. Tokens exceeding $\tau$ are categorized as highly sensitive and thus prioritized for maximal information preservation, minimizing disruption during processing.

## 4 EXPERIMENTS

### 4.1 EXPERIMENTAL SETTINGS

We sample seven tasks from the MileBench benchmark (Song et al., 2024), which is the first benchmark specifically designed to test the long-context multimodal capabilities of MLLMs. MileBench covers a wide range of general scenarios, including temporal multi-image tasks, semantic multi-image tasks, needle-in-a-haystack tasks, and image retrieval tasks. On average, each sample in MileBench contains 15.2 images and 422.3 words.

Table 1: Performance of KV cache compression methods under 20% cache budget. The best results are highlighted in **bold**. Δ denotes the difference to the Full Cache baseline. Note that for MobileVLM-V2-3B on the ImageNeedle task, we don't report its performance because even with full cache, its accuracy is nearly zero. This indicates that the model itself may not be well-suited for this particular task, and thus, the evaluation of effectiveness in this context would not be meaningful.

| Method | ALFRED | IEdit | STD | MMCoQA | CLEVR-C | TextNeedle | ImageNeedle | Average | Δ |
|---|---|---|---|---|---|---|---|---|---|
| *Qwen2.5-VL-7B* | | | | | | | | | |
| Full Cache | 36.92 | 30.16 | 28.13 | 50.50 | 45.45 | 11.56 | 24.38 | 32.44 | - |
| StreamingLLM | 27.61 | 30.43 | 26.85 | 46.00 | 37.00 | 4.38 | 1.88 | 24.88 | -7.57 |
| H2O | 34.31 | 30.91 | 26.63 | 45.50 | **42.49** | 4.69 | 5.00 | 27.08 | -5.36 |
| D2O | 33.59 | 30.56 | 26.16 | 39.50 | 41.58 | 4.69 | 8.75 | 26.40 | -6.04 |
| KVMerge | 27.94 | 31.16 | 27.83 | 44.50 | 37.95 | 9.69 | 15.00 | 27.72 | -4.72 |
| LOOK-M | 34.76 | 30.58 | 25.37 | 39.50 | 40.41 | 2.50 | 3.13 | 25.18 | -7.26 |
| FlowMM | **35.43** | **31.67** | **28.08** | **48.50** | 41.79 | **10.00** | **17.13** | **30.37** | **-2.07** |
| *InternVL2.5-8B* | | | | | | | | | |
| Full Cache | 35.34 | 9.12 | 26.37 | 52.50 | 22.76 | 25.00 | 24.69 | 27.97 | - |
| StreamingLLM | 23.36 | 10.71 | 25.33 | 51.50 | 19.39 | 10.00 | 2.50 | 20.40 | -7.57 |
| H2O | 33.05 | 11.31 | 26.21 | 51.50 | 21.01 | 10.93 | 21.25 | 25.04 | -2.93 |
| D2O | 32.63 | 11.08 | 26.64 | 50.00 | 20.82 | 11.25 | 21.25 | 24.81 | -3.16 |
| KVMerge | 33.61 | 11.42 | 26.33 | 51.50 | 20.47 | 17.63 | 21.45 | 26.06 | -1.91 |
| LOOK-M | 31.81 | 11.18 | **26.82** | 51.00 | 21.21 | 8.25 | 17.15 | 23.92 | -4.05 |
| FlowMM | **34.77** | **11.93** | 26.59 | **52.00** | **22.58** | **23.12** | **23.93** | **27.85** | **-0.12** |
| *MobileVLM-V2-3B* | | | | | | | | | |
| Full Cache | 25.18 | 6.55 | 13.57 | 7.00 | 15.97 | 9.68 | - | 12.99 | - |
| StreamingLLM | 13.73 | 5.82 | 7.65 | 2.50 | 6.55 | 3.12 | - | 6.56 | -6.43 |
| H2O | 24.86 | 6.22 | 10.27 | 3.00 | 14.06 | 2.50 | - | 10.15 | -2.84 |
| D2O | 23.96 | 6.48 | 11.59 | 4.50 | 13.85 | 4.34 | - | 10.79 | -2.20 |
| KVMerge | 24.47 | 6.39 | 11.51 | 4.00 | 14.67 | 5.38 | - | 11.07 | -1.92 |
| LOOK-M | 24.40 | 6.12 | 10.86 | 4.00 | 13.04 | 2.87 | - | 10.22 | -2.78 |
| FlowMM | **25.06** | **6.57** | **12.73** | **5.50** | **15.39** | **8.67** | - | **12.32** | **-0.67** |

To comprehensively evaluate FlowMM, we conduct experiments on several widely-adopted MLLMs: Qwen2.5-VL-7B (Bai et al., 2025), InternVL2.5-8B (Chen et al., 2024b), and MobileVLM-V2-3B (Chu et al., 2024). These models represent diverse architectures, enabling a robust assessment of FlowMM's effectiveness across different model designs. We compare FlowMM against five KV cache compression baselines. StreamingLLM (Xiao et al., 2023b) and H2O (Zhang et al., 2023b) employ eviction-based strategies, while D2O (Wan et al., 2024a) and KVMerge (Wang et al., 2024c) utilize merging-based approaches. All four are text-based KV cache compression methods. Additionally, we compare against LOOK-M (Wan et al., 2024b), a multimodal-specific KV cache merging method.

## 4.2 MAIN RESULT

In Table 1, we present a comparative evaluation of FlowMM against prominent KV cache compression methods in multimodal long-context scenarios. The results highlight FlowMM's efficacy in managing KV cache under strict memory constraints while maintaining competitive task performance. Notably, FlowMM achieves a substantial 80% reduction in memory usage with only a minimal 0.12% average accuracy degradation on InternVL-2.5-8B compared to full cache retention.

Furthermore, FlowMM consistently surpasses eviction-based baselines across most datasets. This advantage is particularly evident in the challenging TextNeedle task, where FlowMM delivers a significant 5.31% accuracy improvement on Qwen2.5-VL-7B. This performance gap underscores a key limitation of eviction methods: their discarding of KV entries inherently leads to context loss, directly contributing to suboptimal model responses. FlowMM also outperforms merging-based approaches. We attribute this superiority to FlowMM's layer-adaptive merging strategy, which dy-

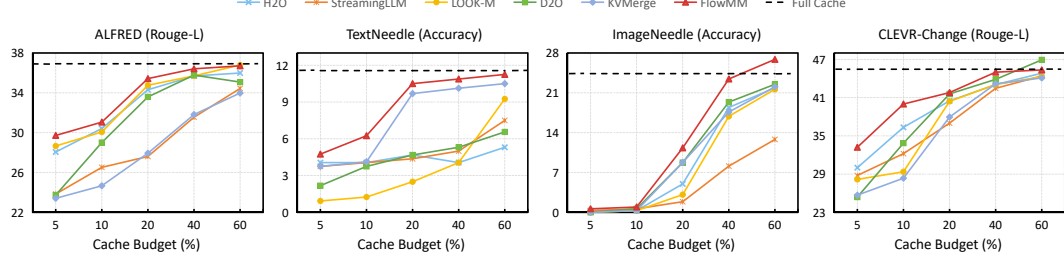

Figure 4: Evaluation results of FlowMM and other KV cache compression methods with varied cache budgets.

namically adjusts merging decisions by identifying cross-modal attention flows. This mechanism effectively prevents modality confusion during merging while fostering deeper semantic relationships across modalities, thereby enhancing the model's capability to comprehend complex multimodal contexts.

## 4.3 Influence of Various Cache Compression Ratios

To validate the effectiveness of FlowMM under varying cache budgets, we conduct experiments on the Qwen2.5-VL-7B model with cache budgets ranging from 5% to 60%. We select four tasks for evaluation: ALFRED, Text Needle In A Haystack, Image Needle In A Haystack, and CLEVR-Change. The results are presented in Figure 4. FlowMM consistently outperform the baseline across all budgets. Notably, in the Text Needle In A Haystack task, FlowMM achieve significantly better performance with a 20% cache budget than the eviction-based method with a 60% cache budget. When the cache budget is below 10%, FlowMM demonstrates a substantial advantage over the baseline, indicating that cross-modal information flow alignment approach effectively retains crucial multimodal contextual information. Moreover, FlowMM achieves performance comparable to full caching with a 40% cache budget and even surpasses full caching in the Image Needle In A Haystack task with a 60% cache budget. We attribute this to FlowMM's dynamic identification of token sensitivity during the merging process, which effectively prevents the dilution of task-specific key contexts and minimizes the excessive merging of task-irrelevant information.

## 4.4 Efficiency Analysis

As shown in Table 2, we evaluate the efficiency of our proposed method. Specifically, we measure decoding speed and GPU memory consumption during inference, comparing configurations with and without our approach. To ensure reliable and robust findings, all tests are conducted using 20 randomly sampled data entries on a single NVIDIA A100 Tensor Core GPU.

Table 2: Model Speed and KV Cache GPU Memory Usage. The best results are highlighted in **bold**.

| Method | Budget | Decoding Latency | GPU Memory |
|---|---|---|---|
| Full Cache | 100% | 29.08 ms/token | 2.06 GiB |
| FlowMM | 50% | 23.04 ms/token | 1.05 GiB |
| | 35% | 19.18 ms/token | 0.74 GiB |
| | 20% | 17.35 ms/token | 0.44 GiB |
| | 5% | **15.81 ms/token** | **0.13 GiB** |

FlowMM demonstrates substantially reduced decoding latency compared to the full-cache model. This advantage is particularly pronounced in long-context tasks, where the efficiency of our method is further enhanced. We further analyze GPU memory utilization under varying

Table 3: Performance under different cross-modal merging threshold $\theta$.

| | 0.1 | 0.2 | 0.3 | 0.4 | 0.5 | 0.6 |
|---|---|---|---|---|---|---|
| TextNeedle | 8.36 | **10.00** | 9.51 | 8.47 | 7.38 | 7.09 |
| ALFRED | 34.69 | 35.11 | **35.43** | 34.78 | 34.92 | 33.61 |

KV cache budgets, with results averaged across inference runs on 20 randomly selected data points. Our findings indicate that the average GPU memory consumption is nearly proportional to the cache budget. Specifically, with a 20% KV cache budget, the memory usage during the decoding phase is reduced by approximately 80% compared to the full cache scenario. This highlights the effectiveness of FlowMM for KV cache compression.

### 4.5 ABLATION STUDY

#### 4.5.1 CROSS-MODAL MERGING THRESHOLD $\theta$.

The cross-modal merging threshold $\theta$ dynamically controls the merging strategy applied at specific layers. To assess its impact, we conduct experiments on Qwen2.5-VL-7B. As presented in Table 3, we observe peak model performance across diverse datasets when the threshold $\theta$ is set between 0.2 and 0.3. Overly low $\theta$ values trigger cross-modal merging too early in the network. This premature fusion occurs before tokens from different modalities have sufficiently interacted, leading to confusion of information and consequently, performance deterioration. Conversely, an excessively high $\theta$ value restricts merging predominantly to within individual modalities throughout most layers. This limitation prevents adequate cross-modal fusion, hindering the model's ability to effectively integrate heterogeneous information and resulting in suboptimal performance.

#### 4.5.2 EFFECTIVENESS OF EACH COMPONENT.

We conduct ablations to validate the necessity of core components in our FlowMM. We evaluate Qwen2.5-VL-7B on three benchmark datasets: TextNeedle, STD, and ALFRED. As shown in Table 4, both cross-modal information flow guidance and sensitivity-adaptive token preservation are critical for performance.

Table 4: Ablation study of the effect of individual module.

| Method | TextNeedle | STD | ALFRED |
|---|---|---|---|
| Full Cache | 11.56 | 28.13 | 36.92 |
| FlowMM | **10.00** | **28.08** | **35.43** |
| w.o. Information Flow Guidance | 5.67 | 26.32 | 33.58 |
| w.o. Sensitivity-Adaptive Matching | 6.32 | 27.14 | 33.75 |
| w.o. both | 3.61 | 25.24 | 31.01 |

Cross-modal information flow quantifies the interaction intensity between heterogeneous modalities. This metric enables adaptive KV cache merging strategies tailored to each layer's specific interaction pattern. As demonstrated in Table 4, removing this adaptive guidance incurs significant performance degradation. The removal of this strategy results in a performance drop, which underscores its efficacy in multimodal long contexts. This finding corroborates our earlier assertion that there are significant differences in cross-modal interaction intensity across different layers of MLLMs. Neglecting these layer-wise differences risks suboptimal multimodal information integration. By allowing the model to dynamically adjust the merging strategy based on the interaction pattern of each layer, cross-modal information flow guidance enables the model to maximize context integration while preserving its inherent cross-modal processing characteristics.

As shown in Table 4, disabling token sensitivity preservation consistently degrades performance across all tasks. This effect is particularly pronounced in the TextNeedle task, where performance drops by 3.68%, thus establishing the effectiveness of our approach. These results underscore the necessity of preserving highly sensitive, task-relevant tokens within multimodal long-context scenarios. Our merging strategy incorporates both token similarity and sensitivity. This dual-pronged approach not only facilitates effective context integration but also safeguards against performance degradation caused by misalignment and dilution of critical information during the merging process.

## 5 CONCLUSION

In this work, we introduce FlowMM, an adaptive framework for cross-modal KV cache merging guided by multimodal information flow. FlowMM dynamically determines cross-modal interaction patterns through layer-wise information flow analysis, enabling layer-specific merging strategies to integrate contextual information. Moreover, our sensitivity-aware token matching jointly assesses token similarity and their task-specific sensitivity, preserving highly sensitive and informative tokens. Extensive experiments demonstrate that FlowMM achieves accuracy comparable to full KV cache utilization while significantly outperforming existing KV cache compression methods across multiple tasks. While this work focuses on image-text modalities, future efforts will explore extending FlowMM to video-audio models, where the longer temporal sequences and higher-dimensional features impose higher memory pressures.

## 6 ETHICS STATEMENT

This work adheres to the ethical principles outlined in the ICLR Code of Ethics, emphasizing responsible stewardship, scientific excellence, and societal well-being. We acknowledge the global stakeholders in machine learning research and strive to ensure our contributions benefit society while minimizing potential harms. Our research upholds high standards of integrity, transparency, and reproducibility, with methods and results reported accurately and honestly. We have carefully considered the broader impacts of our work, including potential risks to privacy, safety, and fairness, and have engaged with relevant domain experts to mitigate unintended consequences. Any data used in this study was handled in accordance with ethical approvals, respecting privacy and confidentiality. We are committed to fostering inclusivity, avoiding discrimination, and ensuring our findings are accessible and socially responsible.

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

# A APPENDIX

## A.1 THE USE OF LARGE LANGUAGE MODELS

In the preparation of this manuscript, LLM is utilized as a general-purpose assist tool for specific tasks. The LLM is employed solely for the following purposes:

- Spelling and Grammar Checking: The LLM is used to identify and correct spelling errors and grammatical inconsistencies, such as verb tense agreement, across the manuscript.
- Sentence Polishing: The LLM provides suggestions for rephrasing sentences to enhance clarity and readability, without altering the original meaning or technical content of the text. All suggestions are reviewed and approved by the authors to ensure alignment with the intended scientific contributions.

The use of the LLM is limited to these auxiliary tasks and does not contribute to the research ideation, methodology, analysis, or core writing of the paper. All scientific content, including ideas, arguments, and conclusions, is developed and written by the authors.

## A.2 DETAILS OF DATASETS

Table 5: Detailed Statistics and Taxonomy of dataset.

| Dataset Abbr. | Task | Data Source | Metric |
|---|---|---|---|
| ALFRED | Conversational Embodied Dialogue | ALFRED (Shridhar et al., 2020) | ROUGE-L |
| IEdit | Visual Relationship Expressing | IEdit (Tan et al., 2019) | ROUGE-L |
| STD | Visual Change Captioning | Spot-the-Diff (Jhamtani & Berg-Kirkpatrick, 2018) | ROUGE-L |
| MMCoQA | Multimodal Dialogue | MMCoQA (Li et al., 2022) | Accuracy |
| CLEVR-C | Visual Change Captioning | CLEVR-Change (Hosseinzadeh & Wang, 2021) | ROUGE-L |
| TextNeedle | Text Needle In A Haystack | TextNeedleInAHaystack | Accuracy |
| ImageNeedle | Image Needle In A Haystack | ImageNeedleInAHaystack | Accuracy |

