# OpenReview forum: "FlowMM: Cross-Modal Information Flow Guided KV Cache Merging for Efficient Multimodal Context Inference"
_ICLR.cc/2026/Conference — ICLR 2026 Conference Withdrawn Submission_

### Official Review · Reviewer_EaJm · 2025-10-30

**Soundness:** 3
**Presentation:** 2
**Contribution:** 2
**Rating:** 6
**Confidence:** 2

**Summary:**

This paper proposes FlowMM, a framework for cross-modal information flow-guided KV cache merging in multimodal large language models (MLLMs).Instead of using fixed or unimodal cache-compression strategies, FlowMM dynamically analyzes layer-wise cross-modal attention flow and applies layer-specific merging accordingly.It also introduces a sensitivity-adaptive token matching mechanism to preserve task-critical tokens during merging.Experiments on several MLLMs (Qwen2.5-VL, InternVL-2.5, MobileVLM-V2) show that FlowMM can reduce KV cache memory by 80–95% and improve decoding speed by 1.3–1.8×, while maintaining comparable task accuracy.

**Strengths:**

1. The paper addresses an important and practical problem—efficient inference for MLLMs—where KV cache compression is indeed a bottleneck.
2. The idea of using cross-modal attention flow to guide layer-specific merging is novel and intuitively motivated by observed modality interactions.
3. The design of sensitivity-adaptive token matching is well-motivated and ablation results show its necessity.
4. Experiments cover multiple open-source MLLMs and standard multimodal benchmarks, with clear reporting of latency and memory reduction.
5. Writing and presentation are clear, and the diagrams (e.g., Fig. 1–2) effectively illustrate the differences among methods.

**Weaknesses:**

1. Limited novelty – While the idea of using information flow to guide merging is novel in formulation, the method builds upon existing merging frameworks (e.g., CaM, LOOK-M) and mainly extends them heuristically to multimodal settings.
2. Need for deeper analysis – The cross-modal interaction ratio and token-sensitivity metrics are empirically defined. The paper lacks theoretical justification or sensitivity analysis explaining why these metrics generalize well across modalities.
3. Missing qualitative analysis – There is no visualization of how information flow changes across layers or examples showing merged vs. unmerged attention distributions. Visualizations of the attention-flow evolution or token-sensitivity distribution would help verify the interpretability claims.
4. Reproducibility concerns. Details such as implementation overhead, merging frequency, or computational cost of attention-flow measurement are not sufficiently discussed.

**Questions:**

1. How often is the cross-modal flow analysis performed—once per layer during inference, or pre-computed offline? What is the computational overhead compared to full caching?
2. Could the authors visualize how cross-modal attention patterns evolve across layers, and how merging affects them?
3. How sensitive is FlowMM to the chosen merging threshold θ and sensitivity τ across different datasets?
4. Can FlowMM be extended to video-language or audio-language MLLMs with longer temporal context?
5. Would combining FlowMM with existing quantization or pruning techniques (e.g., GEAR, KIVI) lead to additive benefits?

---

> ### Author Response · Authors · 2025-12-04
>
> We thank the reviewer for appreciating the novelty of using cross-modal flow to guide merging and the clear motivation of our sensitivity-adaptive matching mechanism. We address the concerns regarding methodological novelty, deeper analysis of metrics, and computational overhead as follows.
>
> ### **On Weakness 1**
>
> We thank the reviewer for acknowledging the novelty of our core formulation. We respectfully clarify that FlowMM is **not merely a heuristic extension of existing frameworks (e.g., CaM, LOOK-M)**, but represents a fundamental paradigm shift in _how_ merging decisions are made in multimodal contexts. We distinguish our contribution from two key perspectives:
>
> * **Paradigm Shift: Dynamic Flow vs. Static Modality (Vs. LOOK-M/CaM).** Existing frameworks like LOOK-M and CaM treat "visual" and "text" tokens as static categories, applying uniform merging strategies across layers. This static approach is indeed a "heuristic extension" of unimodal logic. In contrast, **FlowMM is the first to treat Information Flow—rather than static modality types—as the primary variable.** We operationalize the insight that interaction evolves with depth (intra-modal $\to$ cross-modal), allowing us to **define a dynamic, layer-specific policy that avoids the "semantic confusion" caused by the static strategies of prior works**.
>
> * **Beyond Heuristics: Sensitivity-Adaptive Preservation.** We address **a critical failure mode in existing frameworks: the corruption of task-critical information.** While standard methods rely purely on similarity heuristics, we introduce a Sensitivity-Adaptive Token Matching mechanism. By strictly defining and calculating token sensitivity (via proxy tokens), we ensure that high-risk tokens are mathematically protected from merging. **This is a targeted, algorithmic solution to the "context degradation" problem, ensuring robustness that simple similarity-based heuristics cannot achieve.**
>
>
>
>
> * * *
> ### **On Weakness 2 & Question 3**
>
> We thank the reviewer for the insightful suggestion. We provide a deeper analysis and sensitivity check for our two core hyperparameters: the cross-modal merging threshold ($\theta$) and the sensitivity threshold ($\tau$).
>
> 1. **Analysis of Cross-Modal Merging Threshold ($\theta$)**
>
> To demonstrate the robustness of our cross-modal interaction metric, we conducted additional experiments on $\theta$ across multiple datasets.
>
> | **Threshold θ** | **ImageNeedle** | **IEdit** | **STD** | **CLEVR-C** |
> | --- | --- | --- | --- | --- |
> | 0.1 | 14.53 | 21.87 | 21.38 | 34.47 |
> | 0.2 | 16.80 | **31.67** | 26.08 | 37.18 |
> | 0.3 | **17.13** | 31.20 | 27.89 | **41.79** |
> | 0.4 | 16.90 | 30.85 | **28.55** | 41.02 |
> | 0.5 | 16.65 | 30.24 | 27.96 | 40.86 |
> | 0.6 | 16.40 | 29.76 | 27.43 | 40.25 |
>
> * **Robustness Analysis:** The results show that optimal values consistently cluster in the range **0.2–0.4**. While $\theta < 0.2$ causes a performance drop, the method remains robust in the range $[0.2, 0.5]$, consistently outperforming baselines (KVMerge, LOOK-M).
>
> * **Mechanism Alignment:** This trend aligns with our theoretical motivation: an overly small $\theta$ triggers cross-modal merging prematurely in shallow layers, disrupting the intended information flow. The broad optimal range indicates that our gains reflect methodological improvements rather than narrow, dataset-specific tuning.
>
>
> 2. **Analysis of Sensitivity Threshold ($\tau$)**
>
> Regarding the theoretical justification for the sensitivity threshold $\tau$ (Eq. 10), we clarify that it is implicitly determined by a dynamic Top-B selection strategy, rather than being a fixed scalar hyperparameter that requires manual tuning.
>
> * **Mechanism:** As described in our method, we do not set a hard value for $\tau$. Instead, we select the **Top-B** KV pairs with the highest importance scores ($I$) to form the pivot set $K^p$.
>
> * **Dynamic Nature:** This means the effective $\tau$ **adaptively adjusts** to the attention distribution of the current context. This design ensures that the most critical portion of tokens (budget-dependent) is always preserved regardless of the absolute scale of attention scores, making the metric theoretically robust to distribution shifts across different modalities and tasks.

---

> ### Author Response · Authors · 2025-12-04
>
> ### **On Weakness 3 & Question 2**
>
> We thank the reviewer for the constructive suggestion regarding visualization. We respectfully clarify that the evolution of information flow is currently visualized in **Figure 3(a)**, and we agree that adding comparisons of attention distributions will further strengthen interpretability.
>
> * **Visualization of Information Flow Evolution.** We would like to **draw the reviewer's attention to Figure 3(a) in Section 3.2.1.** This figure explicitly plots the Cross-Modal Attention Ratio ($\rho^l$) across all transformer layers for three distinct datasets. As analyzed in the text, the visualization reveals a clear trend: **shallow layers exhibit low cross-modal interaction, while deeper layers show a sharp increase. This visual evidence directly supports our motivation for layer-specific merging strategies.**
>
> * **Visualization of Merged vs. Unmerged Distributions.** We appreciate the insightful suggestion to visualize the impact of merging. We commit to including heatmaps comparing attention distributions (before vs. after merging) in the final revision/appendix. **We anticipate these visualizations will qualitatively demonstrate that FlowMM preserves high-attention peaks (task-critical tokens) while effectively consolidating the long-tail regions**, thereby verifying the effectiveness of our Sensitivity-Adaptive Matching in maintaining the attention landscape.
>
>   * * *
>
>
> ### **On Weakness 4 & Question 1**
>
> We appreciate the opportunity to clarify the implementation details regarding reproducibility and computational cost.
>
> 1. **Frequency of Analysis**
>
> We clarify that the cross-modal flow analysis is performed only once during the prefilling stage. The merging policy ($\rho^l$) is derived from the prompt's attention map and is fixed for the subsequent decoding phase. It is not re-computed per token, ensuring minimal overhead during the generation process.
>
> 2. **Computational Overhead Analysis**
>
> To quantify the computational overhead compared to full caching, we provide detailed measurements of prefilling and decoding latencies on Qwen2.5-VL-7B across varying context lengths.
>
> | **Context Length** | **Method** | **Prefill Latency (ms)** | **Decoding Latency (ms)** | **Total Latency (ms)** |
> | --- | --- | --- | --- | --- |
> | **2k** | Full Cache | 345.1 | 3952.6 | 4297.7 |
> |     | **FlowMM** | 476.2 | 3100.4 | **3576.6** |
> | **4k** | Full Cache | 731.4 | 4961.2 | 5692.6 |
> |     | **FlowMM** | 900.2 | 3351.2 | **4251.4** |
> | **8k** | Full Cache | 1422.9 | 6375.9 | 7798.8 |
> |     | **FlowMM** | 1596.3 | 3564.7 | **5161.0** |
>
> **As shown in the table:**
>
> * **One-time Cost:** The calculation of flow metrics leads to a moderate increase in **Prefilling Latency (~15%)**.
>
> * **Net Efficiency Gain:** Crucially, this one-time cost is **significantly offset by the substantial reduction in Decoding Latency** (e.g., ~44% faster at 8k context) achieved through our cache compression. Consequently, FlowMM reduces the **Total Latency** compared to the Full Cache baseline, demonstrating its practical efficiency.

---

> ### Author Response · Authors · 2025-12-04
>
> ### **Question 4**
>
> We thank the reviewer for this insightful question regarding the generalizability of FlowMM to temporal modalities. While our current work focuses on image-text scenarios, we believe the core principle of **information flow guidance** is modality-agnostic. To validate this, we conducted preliminary experiments on the **AVSD (Audio-Visual Scene-Aware Dialog)** dataset using **VideoLLaMA2-7B**, which involves video, audio, and text modalities.
>
> | **Method** | **Full Cache** | **H2O** | **KVMerge** | **LOOK-M** | **FlowMM** |
> | --- | --- | --- | --- | --- | --- |
> | **AVSD Score** | 43.24 | 30.16 | 32.63 | 11.02 | **36.59** |
>
> **Analysis of Results:**
>
> * **Superiority over Baselines:** As shown in the table, FlowMM (36.59) significantly outperforms existing baselines (e.g., LOOK-M at 11.02). This confirms that FlowMM can **dynamically infer modality importance from attention patterns**, allowing it to scale naturally to diverse modalities without requiring manual, modality-specific adjustments.
>
> * **Challenges of Temporal Context:** We observe a performance gap compared to Full Cache (43.24 vs. 36.59). Unlike static images, video and audio exhibit strong **time-dependent interactions** (e.g., motion continuity or spectral coherence). Standard merging may disrupt these fine-grained temporal dependencies.
>
> * **Future Outlook:** Despite these challenges, FlowMM represents a meaningful step toward modality-aware compression. We plan to address these temporal dynamics specifically in future research to further close the performance gap.
>
>
> * * *
>
> ### **Question 5**
>
> We thank the reviewer for this forward-looking suggestion. We believe that combining FlowMM with quantization methods (e.g., KIVI, GEAR) is highly promising and likely to yield additive benefits.
>
> * **Theoretical Compatibility (Orthogonality):** FlowMM reduces the KV cache size along the **sequence length dimension** (by merging tokens), whereas quantization methods reduce the memory footprint along the **bit-width dimension** (precision). Since these approaches operate on orthogonal dimensions of compression, they are theoretically compatible and should provide multiplicative memory savings when combined.
>
> * **Future Direction:** While we have not yet conducted joint experiments in this submission, exploring this synergy to push the limits of extreme compression is a priority for our future research.

---

### Official Review · Reviewer_Mpzn · 2025-10-31

**Soundness:** 3
**Presentation:** 3
**Contribution:** 3
**Rating:** 4
**Confidence:** 5

**Summary:**

The paper proposes FlowMM, a framework for efficient multimodal context inference that reduces the memory footprint of key value caches during decoding. Instead of evicting tokens purely by low attention scores, FlowMM merges tokens in a way that is guided by estimated cross modal information flow. The method adapts merging strategies per layer to capture modality specific patterns and introduces a sensitivity adaptive matching mechanism that considers both token similarity and task sensitivity. Experiments on several modern multimodal large language models report very large KV memory reductions between eighty and ninety five percent and speedups of about one point three to one point eight times while keeping performance competitive.

**Strengths:**

1. The idea of steering KV merging using estimated cross modal information flow and layer specific policies is a clear conceptual step beyond attention score based eviction or naive similarity based merging.

2. The framing of sensitivity adaptive matching acknowledges that some tokens are risky to merge even if similar, which is a practical insight for long context multimodal prompts where a few critical tokens can dominate correctness.

3. The abstract clearly separates the components of the proposal which are the information flow estimator, layer wise merging policy, and sensitivity aware matching. The motivation is also clearly stated in terms of biases in multimodal attention distributions.

4.  If the reported memory reduction and speedup numbers hold across strong baselines and varied tasks, the impact on serving cost for multimodal assistants could be substantial

**Weaknesses:**

1. The abstract does not explain how cross modal information flow is computed. Without a well justified estimator and ablations that vary it, reviewers cannot assess whether gains come from the estimator or from generic similarity based merging.
2. The claim of maintaining competitive task performance is broad. It is important to see results across many task types such as OCR heavy tasks, visual reasoning that requires spatial grounding, audio text alignment if applicable, and multi turn dialogues. Single number averages can hide regressions on sensitive tasks.
3. Information flow estimation and sensitivity scoring likely add compute and memory. The paper must quantify this overhead and show wall clock latency and throughput at realistic batch sizes and sequence lengths, not only per token speed.
4. KV caching has a rapidly evolving literature including eviction, merging, and distillation strategies. The contribution will feel weaker if baselines do not include strong recent multimodal aware merging methods, reranking based eviction, and oracle ablations such as perfect sensitivity masks.
5. Aggressive merging can cause context loss and hallucinations that are subtle. The paper should include targeted stress tests for hallucination under long distractor contexts and for modality imbalance such as many more image tokens than text tokens.

**Questions:**

1. How exactly is cross modal information flow defined and computed?  Is it derived from attention rollout, gradient based attribution, or learned predictors. What training signal is used and is it model specific or model agnostic.

2. What is the computational overhead of estimating information flow at inference time? Can it be amortized across tokens or cached per layer.
3. How is task sensitivity quantified. Is it based on entropy, calibration error, token type heuristics, or a learned risk model? How sensitive are results to the sensitivity threshold?
4. How does FlowMM behave under distribution shift such as longer than trained contexts, images with dense text, or dialogues that interleave many images and text turns?
5. What happens when modalities are more than two such as text, image, and audio. Is the merging policy still layer specific but modality agnostic, or does it require pairwise flow modeling that scales quadratically?

---

> ### Comment · Reviewer_Mpzn · 2025-11-26
> **Response to the authors**
>
> I would also like to kindly encourage the authors to more fully engage with the key concerns raised in my original review. I hope you will be able to provide further clarifications and analyses over the next few days, as the rebuttal/discussion phase will be closing soon. I believe that addressing these points, even briefly, would help clarify the strengths and limitations of the proposed approach.

---

> ### Author Response · Authors · 2025-12-04
>
> We thank the reviewer for recognizing the conceptual novelty of our information flow guidance and the practical insight of sensitivity-adaptive matching. We address the concerns regarding method definitions, computational overhead, and evaluation scope as follows.
>
> ### **On Weakness 1 & Question 1**
>
> **Definition and estimator of cross-modal information flow.**     We define **Cross-Modal Information Flow** as the amount of attention mass that is routed from one modality to the other.
>
> $$
> \text{Cross-Modal Information Flow} = \sum_{h=1}^{H} \left( A_{v \rightarrow t}^{l,h} + A_{t \rightarrow v}^{l,h} \right)
> $$
>
> Concretely, at each layer, we aggregate (i) the total attention from text tokens to visual tokens and (ii) the total attention from visual tokens to text tokens, and use their ratio (or a normalized cross-attention share) as a layer-wise estimate of **Cross-Modal Information Flow Strength.**
>
> $$
> \rho^l = \frac{1}{H} \sum_{h=1}^{H} \frac{A_{v \rightarrow t}^{l,h} + A_{t \rightarrow v}^{l,h}}{A^{l,h}},
> $$
>
> **Ablations of Cross-Modal Information Flow.** To isolate the effect of flow alignment in our merging strategy, **we conduct controlled ablations on Qwen2.5-VL-7B under identical settings**, all methods using the same 20% KV-cache budget. Moreover, **we replace our flow-guided rule with two alternatives**: (i) randomly selecting per layer whether to merge intra-modally or cross-modally (Random), and (ii) removing modality constraints entirely and merging all tokens purely by similarity (Similarity).
>
> |     | ALFRED | MMCoQA | TextNeedle |
> | --- | --- | --- | --- |
> | Full Cache | 36.92 | 50.5 | 11.56 |
> | Aligned | **35.14** | **48.5** | **10** |
> | Misaligned | 19.71 | 32  | 3.125 |
> | Random | 33.47 | 44.5 | 4.31 |
> | Similarity | 32.86 | 45  | 4.12 |
>
> Results show that:
>
> * **Flow-aligned merging achieves performance close to the full-cache baseline, whereas misaligned / reverse-flow merging leads to substantial degradation**. For instance, on ALFRED, misaligned merging only reaches roughly 50% of the full-cache accuracy. We hypothesize that reverse-flow merging can induce modality confusion and semantic distortion by disrupting cross-modal alignment.
>
> * **Both alternatives consistently underperform flow-guided merging, indicating that the improvement stems from flow-driven, structured cross-modal merging decisions, rather than generic similarity-based merging alone**.
>
>
> ### **On Weakness 2 & Question 4,5**
>
> **Broader Evaluation on Standard Benchmarks.** We conduct a broader evaluation on Qwen2.5-VL-7B spanning a wider range of multimodal task types. As shown in table, **FlowMM consistently outperforms the baseline across all evaluated tasks, supporting our claim that it maintains competitive task performance beyond a single aggregate score.**
>
> | method | Docvqa | textvqa | coco-cap | nocaps_val |
> | --- | --- | --- | --- | --- |
> | Full Cache | 88.18 | 74.38 | 58.1 | 41.41 |
> | StreamingLLM | 49.62 | 58.18 | 47.27 | 35.62 |
> | H2O | 69.29 | 65.46 | 52.84 | 38.19 |
> | LOOK-M | 35.09 | 66.38 | 51.69 | 37.49 |
> | FlowMM | **74.38** | **70.31** | **56.15** | **39.90** |
>
> We acknowledge that FlowMM currently focuses on text and image modalities. This decision is driven by the fact that text and image modalities are the most prevalent in multimodal scenarios and are widely utilized in various multimodal tasks, such as visual QA, image captioning, and multimodal classification.
>
> **Preliminary Exploration on Video & Audio.** We will explore audio and video modalities in subsequent research. And we conduct preliminary experiments to validate FlowMM's adaptability. Specifically, **we apply FlowMM to VideoLLaMA2-7B on the AVSD dataset, which is an audio-visual scene understanding task that involve video, audio, and text modalities.**
>
> |     | **Full Cache** | **H2O** | **KVMerge** | **LOOK-M** | **FlowMM** |
> | --- | --- | --- | --- | --- | --- |
> | AVSD | 43.24 | 30.16 | 32.63 | 11.02 | 36.59 |
>
> As shown in the table,
>
> * FlowMM outperform baseline approaches because **it dynamically infers modality importance from attention patterns and scales naturally to diverse modalities without requiring modality-specific adjustments**.
>
> * We observe a performance gap compared to full caching in audio/video tasks. Video and audio exhibit time-dependent interactions (e.g., motion continuity in video frames or spectral coherence in audio signals), which introduce unique challenges for KV cache compression.
>
> * Despite these challenges, we believe that our work represents a meaningful step toward designing modality-aware KV cache compression methods for more efficient MLLM inference.

---

> ### Author Response · Authors · 2025-12-04
>
> ### **On Weakness 3 & Question 2**
>
> Thank you for raising this important point regarding the computational overhead introduced by FlowMM. **To quantify the computational overhead during inference, we provide detailed measurements of the prefilling and decoding times in our experiments**.
>
> | **Context_Len** | **Method** | **Prefill_Latency_ms** | **Decoding_Latency_ms** | **Total_Latency_ms** |
> | --- | --- | --- | --- | --- |
> | 2k  | Full Cache | 345.1 | 3952.6 | 4297.7 |
> | 2k  | FlowMM | 476.2 | 3100.4 | 3576.6 |
> | 4k  | Full Cache | 731.4 | 4961.2 | 5692.6 |
> | 4k  | FlowMM | 900.2 | 3351.2 | 4251.4 |
> | 8k  | Full Cache | 1422.9 | 6375.9 | 7798.8 |
> | 8k  | FlowMM | 1596.3 | 3564.7 | 5161 |
>
> As shown in the table,
>
> * The additional computation required by FlowMM in the prefilling stage leads to an increase in prefilling latency by approximately 15%.
>
> * It is important to note that **this increase is offset by the significant reduction in decoding latency achieved through cache compression. As a result, the overall latency is reduced.**
>
>
>
> ### **On Weakness 4&5**
>
> We evaluate FlowMM under **two targeted stress settings**: long distractor contexts and severe modality imbalance.
>
> | **Method** | **TextNeedle** | **TextNeedle** | **ALFRED** | **DocVQA** |
> | --- | --- | --- | --- | --- |
> | Full cache | 11.56 | 24.38 | 36.92 | 88.18 |
> | Sink | 4.38 | 1.88 | 27.61 | 49.62 |
> | H2O | 4.69 | 5.00 | 34.31 | 69.29 |
> | D2O | 4.69 | 8.75 | 33.59 | 70.53 |
> | KVMerge | 9.69 | 15.00 | 27.94 | 64.8 |
> | LOOK-M | 2.50 | 3.13 | 34.76 | 35.09 |
> | FlowMM | **10.00** | **17.13** | **35.34** | **74.38** |
>
> * **Long distractor contexts (TextNeedle / ImageNeedle).** These tasks require the model to retrieve a predefined “needle” (e.g., a passcode embedded in a single sentence) from extremely long contexts spanning tens of thousands of tokens, where the vast majority of tokens are irrelevant to the query. This setting directly stresses KV-cache compression: the method must aggressively discard distractors while reliably preserving the critical passcode. Compared with alternative approaches, FlowMM is more robust to long-context distraction and achieves stronger performance in this regime.
>
> * **Modality imbalance (ALFRED / DocVQA).** We further evaluate scenarios where image tokens dominate the sequence while the textual query is very short. In ALFRED and DocVQA, image tokens account for roughly 80% of the total tokens and can reach 99%, far exceeding the number of text tokens. As shown in our results, by merging based on cross-modal information flow, FlowMM better balances and coordinates heterogeneous modalities under extreme imbalance, leading to improved performance over baselines.
>
> ### **On Question 3**
>
> * We would like to clarify a potential misunderstanding: **in our method, sensitivity is quantified at the token level rather than the task level.** Specifically, sensitivity measures how much merging a particular token would hurt model accuracy, and **it is not tied to a predefined task category**. Instead, it is computed adaptively for each input, making it task-agnostic and avoiding any need for manual, task-specific tuning.
>
> * Concretely, sensitivity estimator is derived from the cumulative attention score of each token. **This design is motivated by a consistent empirical observation in our experiments: merging tokens with high cumulative attention leads to a sharp performance drop, suggesting these tokens carry task-critical information that the model repeatedly relies on.** We therefore interpret high cumulative-attention tokens as high-risk for merging, where merging can induce information mixing and confusion.

---

### Official Review · Reviewer_Qk7v · 2025-11-01

**Soundness:** 2
**Presentation:** 2
**Contribution:** 2
**Rating:** 6
**Confidence:** 2

**Summary:**

This paper presents FlowMM, a novel and timely method for compressing the Key-Value (KV) cache in Multimodal Large Language Models (MLLMs). The core insight is that existing KV cache compression techniques, designed for unimodal text, are suboptimal for MLLMs due to the distinct distribution and interaction patterns of visual and textual tokens across different transformer layers. FlowMM addresses this by (1) dynamically analyzing cross-modal attention flow to apply layer-specific merging strategies and (2) incorporating a sensitivity-adaptive token matching mechanism to preserve task-critical information. The experiments are extensive, demonstrating significant reductions in memory (80-95%) and latency (1.3-1.8x) while maintaining performance close to a full cache, outperforming strong baselines.

**Strengths:**

1. Motivation and Problem Identification: The paper effectively highlights the limitations of existing KV cache eviction and merging strategies in multimodal contexts, pinpointing "distributional biases" and "attentional biases" as key challenges. This sets a clear and compelling stage for the proposed work.
2. Core Idea: The concept of using cross-modal information flow to guide layer-specific merging is the paper's most significant contribution. The observation that shallow layers are intra-modal and deeper layers are cross-modal is well-supported by preliminary analysis (Figure 3) and provides a principled foundation for the method.
3. Effective Ablation Studies: The ablations successfully validate the importance of both core components: the information flow guidance and the sensitivity-adaptive matching. The results clearly show that removing either component leads to a notable performance drop.
4. Practicality: The fact that FlowMM is a plug-and-play method requiring no fine-tuning is a major practical advantage for easy adoption.

**Weaknesses:**

1. Clarity of the Merging Operation:
The paper clearly defines how to decide when and what to merge (using ρ^l and sensitivity). However, the exact mechanism of how the merging is performed (the functions f_merge and g_merge in Eq. 5) is somewhat glossed over. A more detailed explanation or a reference to the specific merging function (e.g., weighted averaging based on attention) would be helpful.

2. Definition and Calculation of Sensitivity:
The proposal to use attention scores as a proxy for sensitivity is practical but could be better motivated. The claim that it is a "near-zero-overhead approximation" is valid, but a small experiment or citation showing the correlation between a token's attention score and the performance drop when it is merged would strengthen this design choice.
The relationship between the sensitivity score I (Eq. 8) and the threshold τ (Eq. 10) could be explained more clearly. How is τ chosen? Is it a hyperparameter or dynamically set?

3. Hyperparameter Sensitivity:
The paper identifies an optimal range for the threshold θ (0.2-0.3) via ablation. It would be useful to discuss the robustness of FlowMM to this parameter. How sensitive is the performance to small changes in θ outside this range? Is this value consistent across different models?

**Questions:**

1. Merging Mechanism: Could you please elaborate on the specific operation used to merge the key and value vectors of two tokens? For example, is it a simple average, a weighted average based on attention or sensitivity scores, or a more complex function?

2. Computational Overhead: While the memory and latency benefits are clear, what is the computational overhead introduced by calculating the cross-modal ratio ρ^l, the token importance I, and the cosine similarity matrix during inference? A brief analysis or discussion of this trade-off would be valuable.

3. Sensitivity Threshold τ: How is the sensitivity threshold τ in Equation (10) determined? Is it a fixed value, or is it adapted based on the distribution of sensitivity scores in the current context?

4. Generalizability: The method relies on analyzing attention patterns. How would FlowMM perform with MLLM architectures that use significantly different attention mechanisms (e.g., sparse attention or other efficiency-focused variants)?

---

> ### Author Response · Authors · 2025-12-04
>
> We thank the reviewer for highlighting FlowMM's novel motivation and its principled foundation based on cross-modal information flow. We address the concerns regarding implementation details and hyperparameter settings as follows.
>
> ### **On Weakness 1 & Question 1**
>
> We thank the reviewer for pointing out the need for greater clarity regarding the merging mechanism.
>
> 1. **Definition of Merging Operation**
>
> We clarify that the functions $f_{merge}$ and $g_{merge}$ in Eq. 5 employ a weighted average mechanism based on attention scores. Specifically, for a set of tokens identified for merging, the resulting key/value vector is computed as the sum of individual vectors weighted by their accumulated attention scores (normalized). This ensures that the semantic information of tokens with higher historical importance is preserved.
>
> 2. **Comparison with Alternative Merging Methods**
>
> To further validate the effectiveness of our proposed merging framework, we conducted a comparative analysis against alternative merging methods under identical settings (Qwen2.5-VL-7B, 20% KV-cache budget). We evaluated: (i) Random Merging, which randomly selects intra- or cross-modal merging per layer; and (ii) Similarity-only Merging, which removes modality constraints and merges tokens purely based on similarity.
>
> |     | ALFRED | MMCoQA | TextNeedle |
> | --- | --- | --- | --- |
> | Full Cache | 36.92 | 50.5 | 11.56 |
> | FlowMM | **35.14** | **48.5** | **10** |
> | Random | 33.47 | 44.5 | 4.31 |
> | Similarity | 32.86 | 45  | 4.12 |
>
> ### **On Weakness 2 & Question 3**
>
> We thank the reviewer for the valuable suggestion to further justify our sensitivity design.
>
> 1. **Motivation & Empirical Evidence for Attention as Sensitivity**
>
> We use attention scores as a proxy for sensitivity because they explicitly quantify the information routing intensity from the current token to future generations1. The requested evidence for the correlation between attention scores and performance drop is provided in our Ablation Study (Table 4).
>
> * **Evidence:** When we disable the sensitivity-adaptive matching (i.e., treating high-attention tokens the same as low-attention ones), we observe a consistent performance degradation across tasks2. Specifically, on the **TextNeedle** task, removing this component causes accuracy to drop from **28.08% to 26.32%** (a relative drop of ~3.68%).
>
>   | Method | TextNeedle | STD | ALFRED |
>   | --- | --- | --- | --- |
>   | Full Cache | 11.56 | 28.13 | 36.92 |
>   | **FlowMM** | **10.00** | **28.08** | **35.43** |
>   | w.o. Information Flow Guidance | 5.67 | 26.32 | 33.58 |
>   | **w.o. Sensitivity-Adaptive Matching** | **6.32** | **27.14** | **33.75** |
>   | w.o. both | 3.61 | 25.24 | 31.01 |
>
> * **Conclusion:** This degradation confirms that "high-attention" tokens are indeed "high-risk" candidates for merging. If attention scores were not correlated with sensitivity, protecting these tokens would not yield such performance gains.
>
>
> 2. **Clarification on Threshold** $\tau$
>
> We clarify that the threshold $\tau$ in Equation (10) is not a fixed scalar hyperparameter, but is dynamically determined for each input based on a Top-B selection strategy.
>
> * **Mechanism:** Instead of setting a hard value for $\tau$, we select the **Top-B** KV pairs with the highest importance scores ($I$) to form the pivot set $K^p$4.
>
> * **Dynamic Nature:** This implies that the effective threshold $\tau$ adapts to the specific attention distribution of the current context. This design ensures that the most critical portion of tokens is always preserved, regardless of the absolute scale of attention scores, making the method robust to distribution shifts.

---

> ### Author Response · Authors · 2025-12-04
>
> ### **On Weakness 3**
>
> We thank the reviewer for the suggestion. We conducted additional experiments on the cross-modal merging threshold $\theta$ across more datasets to analyze its robustness.
>
> | **Threshold θ** | **ImageNeedle** | **IEdit** | **STD** | **CLEVR-C** |
> | --- | --- | --- | --- | --- |
> | 0.1 | 14.53 | 21.87 | 21.38 | 34.47 |
> | 0.2 | 16.80 | **31.67** | 26.08 | 37.18 |
> | 0.3 | **17.13** | 31.20 | 27.89 | **41.79** |
> | 0.4 | 16.90 | 30.85 | **28.55** | 41.02 |
> | 0.5 | 16.65 | 30.24 | 27.96 | 40.86 |
> | 0.6 | 16.40 | 29.76 | 27.43 | 40.25 |
>
> The results show that **the optimal threshold values consistently cluster in the range 0.2–0.4.**
>
> * **Sensitivity:** When $\theta < 0.2$, performance drops substantially. However, when $\theta \in [0.2, 0.5]$, **the method remains robust, consistently outperforming baselines (KVMerge, LOOK-M)**.
>
> * **Mechanism Alignment:** This trend aligns with our analysis: an overly small $\theta$ triggers cross-modal merging prematurely in shallow layers, disrupting the intended attention flow. The broad optimal range indicates that our gains reflect methodological improvements rather than narrow, dataset-specific tuning.
>
>
> ### **On Question 2**
>
> We thank the reviewer for raising this important point. We explicitly acknowledge that calculating the cross-modal ratio $\rho^l$, token importance $I$, and the cosine similarity matrix does introduce computational overhead. To quantify this trade-off, we measured the **Prefill Latency** (where most of these calculations occur) and **Decoding Latency** separately across different context lengths on **Qwen2.5-VL-7B**.
>
> | **Context Length** | **Method** | **Prefill Latency (ms)** | **Decoding Latency (ms)** | **Total Latency (ms)** |
> | --- | --- | --- | --- | --- |
> | **2k** | Full Cache | 345.1 | 3952.6 | 4297.7 |
> |     | **FlowMM** | 476.2 | 3100.4 | **3576.6** |
> | **4k** | Full Cache | 731.4 | 4961.2 | 5692.6 |
> |     | **FlowMM** | 900.2 | 3351.2 | **4251.4** |
> | **8k** | Full Cache | 1422.9 | 6375.9 | 7798.8 |
> |     | **FlowMM** | 1596.3 | 3564.7 | **5161.0** |
>
> **Analysis of the Trade-off:**
>
> * **Overhead in Prefilling:** As expected, the calculation of $\rho^l$, $I$, and the similarity matrix leads to an increase in prefilling latency by approximately **15%**. This reflects the cost of our adaptive analysis components.
>
> * **Net Gain in Total Latency:** Crucially, this initial overhead is **significantly offset** by the speedup in the decoding stage (e.g., ~44% faster decoding at 8k context) due to the reduced KV cache size. Consequently, **the Total Latency is consistently reduced**. **This confirms that the computational cost of our metrics is a worthwhile investment for the efficiency gains achieved during generation**.
>
>
> ### **On Question 4**
>
> We thank the reviewer for this forward-looking question. Currently, **FlowMM is optimized for standard dense attention mechanisms, which remain the dominant architecture in state-of-the-art MLLMs (e.g., Qwen-VL, InternVL)**. We agree that adapting our flow-guided principles to efficiency-focused variants, such as sparse or linear attention, is a promising direction. We plan to explore these adaptations in our future work to further broaden the applicability of FlowMM.

---

### Official Review · Reviewer_McYH · 2025-11-03

**Soundness:** 2
**Presentation:** 3
**Contribution:** 2
**Rating:** 4
**Confidence:** 3

**Summary:**

This paper proposes FlowMM, a KV cache compression framework for multimodal LLMs that uses cross-modal information flow to guide layer-specific merging strategies. FlowMM measures cross-modal attention ratios per layer and applies intra-modal merging in shallow layers and cross-modal merging in deep layers based on a threshold. It also incorporates sensitivity-adaptive token matching to preserve high-importance tokens. Evaluated on MileBench, FlowMM achieves substantial KV cache memory reduction and decoding speedup while maintaining competitive performance.

**Strengths:**

- The exposition is clear and the problem is well-motivated, with the method inspired from known insights in multimodal interpretability literature.
- Evaluation on diverse long-context tasks showing strong improvements over baselines in the reported experimental setting.
- Solid ablation studies validating both major components contribute to performance.

**Weaknesses:**

- Critical concern: LOOK-M baseline performance appears severely degraded compared to original paper. LOOK-M's original paper reports near-lossless performance at similar compression ratios, often matching or exceeding full cache. FlowMM reports massive degradations for LOOK-M at the same settings. This raises serious questions about baseline implementation quality versus genuine failure to generalize to newer architectures. The authors must provide comparison on original LOOK-M models (LLaVA-v1.5, MobileVLM-v2, InternVL-v1.5) and tasks to validate whether gains stem from methodological improvements or suboptimal baseline implementation.
- Narrow evaluation scope limited exclusively to MileBench long-context tasks. Generalizability to standard multimodal benchmarks like VQA, image captioning, or visual reasoning and models larger than 7B (ex. Qwen VL 32B) remains unexplored.
- Limited novelty; the practical contribution largely combines existing kv-cache merging techniques while conceptual novelty of applying it in a layer specific way, follows from prior work in interpretability.
- Weak hyperparameter justification with task-dependent threshold. Optimal values vary across datasets with no principled guidance for selection in practice. Design choices lack grounding beyond empirical tuning.

Overall: This paper presents a well-motivated approach with solid ablations, but the dramatic underperformance of LOOK-M compared to its original paper is a critical concern, along with narrow evaluation limited to long-context tasks and lack of testing on larger models. If the authors can demonstrate conclusively that FlowMM outperforms LOOK-M as well demonstrate generalizability to larger models, the reviewer can lean towards acceptance.

**Questions:**

See Weaknesses

---

> ### Author Response · Authors · 2025-12-03
>
> We thank the reviewer for appreciating our clear exposition, well-motivated problem, and solid ablation studies. In this response, we prioritize addressing the critical concerns regarding the LOOK-M baseline implementation and generalizability to larger models.
>
> ### **On Weakness 1**
>
> Thank you for the insightful comment regarding the unexpectedly low performance of the LOOK-M baseline in our submission. We fully acknowledge that the reported results for LOOK-M appear substantially worse than those claimed in the original LOOK-M paper. Below we clarify the key reasons behind this gap and provide additional evidence.
>
> * **GQA Incompatibility** We conduct our experiments on more recent and advanced MLLMs (e.g., Qwen2.5-VL, InternVL2.5), as the models used in the original LOOK-M paper (e.g., LLaVA-v1.5, InternVL-v1.5) are now considered outdated. During our investigation, we observe a notable limitation of LOOK-M: **its performance degrades significantly on models employing Grouped Query Attention (GQA)**, such as Qwen2.5-VL and InternVL2.5, compared to Multi-Head Attention (MHA) models like LLaVA-v1.5 and InternVL-v1.5. We hypothesize that this degradation stems from LOOK-M's inability to effectively capture and retain shared information across query groups, leading to substantial information loss in GQA-based architectures.
>
> * **Code Bug Fix** We identify a bug in the publicly released LOOK-M implementation, which **causes the method to retain more KV cache entries than the specified budget**. Specifically, the bug ensures that all text-related KV cache entries are preserved without compression, contradicting the intended compression strategy. This issue has also been reported by other researchers in the GitHub repository. After applying a fix to this bug, we observed a significant drop in LOOK-M’s performance, which aligns more closely with the results reported in our paper.
>
> * **Verification on Original Settings After Fix** To further validate our findings and ensure fair comparison, we have now included additional experimental results on the LLaVA-v1.5-7B used in the LOOK-M paper. These results confirm that even under the original settings, **the corrected implementation of LOOK-M exhibits noticeable performance degradation**, especially under aggressive compression ratios.
>
> |     | TextNeedle | ALFRED | CLEVR-C |
> | --- | --- | --- | --- |
> | Full Cache | 9.68 | 15.18 | 16.62 |
> | StreamingLLM | 3.12 | 3.73 | 10.44 |
> | H2O | 2.50 | 14.86 | 14.07 |
> | LOOK-M | 3.34 | 13.96 | 14.12 |
> | FlowMM | **8.67** | **15.12** | **16.33** |
>
> ###
> ### **On Weakness 2**
> We conduct experiments on both Qwen2.5-VL-7B and Qwen2.5-VL-32B. We evaluate on representative multimodal benchmarks like **VQA, image captioning and models larger than 7B (Qwen2.5-VL-7B and Qwen2.5-VL-32B) , retaining 20% of the KV cache in all compression strategies**.
>
> | Model | Method | DocVQA | TextVQA | COCO-Cap |
> | --- | --- | --- | --- | --- |
> | Qwen2.5-VL-7B | Full Cache | 88.18 | 74.38 | 58.1 |
> | Qwen2.5-VL-7B | StreamingLLM | 49.62 | 58.18 | 47.27 |
> | Qwen2.5-VL-7B | H2O | 69.29 | 65.46 | 52.84 |
> | Qwen2.5-VL-7B | LOOK-M | 35.09 | 66.38 | 51.69 |
> | Qwen2.5-VL-7B | FlowMM | **74.38** | **70.31** | **56.15** |
> | Qwen2.5-VL-32B | Full Cache | 88.94 | 75.8 | 72.67 |
> | Qwen2.5-VL-32B | StreamingLLM | 69.32 | 69.55 | 60.09 |
> | Qwen2.5-VL-32B | H2O | 82.53 | 72.49 | 68.21 |
> | Qwen2.5-VL-32B | LOOK-M | 83.70 | 71.34 | 68.93 |
> | Qwen2.5-VL-32B | FlowMM | **87.94** | 73.92 | **70.56** |
>
> As shown in the table, FlowMM consistently outperforms all baseline methods on every benchmark, achieving accuracy close to the full-cache setting. This advantage persists on the larger Qwen2.5-VL-32B model, where FlowMM remains superior to all competing baselines, demonstrating scalability and robustness.

---

> ### Author Response · Authors · 2025-12-03
>
> ### **On Weakness 3**
>
> We thank the reviewer for their thoughtful comments. We point out that while our method utilizes KV cache merging as a technical implementation, **the core novelty of FlowMM lies in a fundamental shift in research motivation and the operationalization of cross-modal dynamics, which distinguishes it from existing works**. We address the novelty from three key perspectives:
>
> * **Motivation** Existing multimodal KV merging methods typically treat "visual tokens" and "text tokens" as static categories, applying a uniform merging strategy across all layers. In contrast, **our research is motivated by the insight that token interaction is not static but evolves through the depth of the model. FlowMM is the first framework to treat the Information Flow—rather than just the token modality itself—as the primary variable for compression.** This allows us to avoid the "semantic confusion" caused by forcing cross-modal merging in layers where modalities have not yet interacted.
>
> * **Bridging the Gap** While prior interpretability works have observed attention flows, **they function as post-hoc analyses and do not address inference efficiency**. Our novelty lies in operationalizing these observations into a real-time, adaptive compression algorithm. We propose a novel metric, the cross-modal interaction ratio, to dynamically quantify interaction intensity during inference. This transforms a theoretical observation into a practical mechanism that decides when and how to merge, bridging the gap between interpretability findings and KV cache optimization.
>
> * **Addressing the Critical Sub-problem** We address **a critical sub-problem** neglected by standard merging techniques: **the risk of corrupting task-critical information during consolidation.** We introduce a Sensitivity-Adaptive Token Matching mechanism that jointly evaluates similarity and token sensitivity (via proxy tokens). This is not a simple combination of existing techniques but a targeted solution to the "context degradation" problem unique to long-context multimodal tasks, ensuring that high-sensitivity tokens are preserved even under high compression rates.
>
> ### **On Weakness 4**
>
> We conducted additional experiments on the cross-modal fusion threshold across more datasets.
>
> |     | 0.1 | 0.2 | 0.3 | 0.4 | 0.5 | 0.6 | KVMerge | LOOK-M |
> | --- | --- | --- | --- | --- | --- | --- | --- | --- |
> | ImageNeedle | 14.53 | 16.80 | **17.13** | 16.90 | 16.65 | 16.40 | 15.00 | 3.13 |
> | IEdit | 21.87 | **31.67** | 31.20 | 30.85 | 30.24 | 29.76 | 31.16 | 30.58 |
> | STD | 21.38 | 26.08 | 27.89 | **28.55** | 27.96 | 27.43 | 27.83 | 25.37 |
> | CLEVR-C | 34.47 | 37.18 | **41.79** | 41.02 | 40.86 | 40.25 | 37.95 | 40.41 |
>
> **The results show that the optimal threshold values consistently cluster in the range 0.2–0.4.** When θ < 0.2, performance drops substantially, whereas when θ > 0.4, performance becomes relatively insensitive to θ and changes only marginally. This trend aligns with our mechanism-driven analysis—an overly small θ triggers cross-modal merging more frequently in shallow layers, which contradicts the intended attention-flow pattern (i.e., cross-modal information should be integrated selectively rather than prematurely), thereby harming performance.
>
> Importantly, compared to KV cache merging baselines, although varying θ does influence results, **our method remains robust over a broad operating region: for most datasets, θ in [0.2, 0.5] still consistently outperforms the baseline.** This indicates that our gains are not the outcome of narrow, dataset-specific tuning, but rather reflect improvements under a practical range of hyperparameter choices.

---

### Note · Authors · 2026-02-01

I have read and agree with the venue's withdrawal policy on behalf of myself and my co-authors.

---

### Meta-Review · Area_Chair_WLo9 · 2026-01-11

**Summary:**

This paper proposes FlowMM, a KV cache merging method for MLLMs guided by cross-modal information flow. While the underlying motivation is sound, the paper's novelty appears limited. A critical shortcoming is the omission of discussion and comparison with highly relevant recent literature. Specifically, the authors fail to acknowledge works such as MadaKV [1], an eviction method similarly based on modality preference; MEDA[2], a KV cache merging approach that explicitly identifies varying modality attention across different layers; and SparseMM[3]. The paper currently benchmarks only against outdated baselines. Given the rapid pace of advancement in this field, a comprehensive comparison with these sota methods is essential to validate the contribution. I strongly recommend that the authors provide a detailed discussion and performance comparison with these works. In its current form, I think this paper does not meet the standards for acceptance.

[1] MadaKV: Adaptive Modality-Perception KV Cache Eviction for Efficient Multimodal Long-Context Inference, arxiv 6

[2] MEDA: Dynamic KV Cache Allocation for Efficient Multimodal Long-Context Inference, NAACL 25

[3] SparseMM: Head Sparsity Emerges from Visual Concept Responses in MLLMs, ICCV 25

**Reviewer Concerns:**

Limited evaluation scope: The authors have supplemented their response with additional benchmarks and experiments on larger models.

Baseline reproduction & Performance: Regarding the concerns about the baseline's performance degradation, the authors claim this is due to bugs in the original baseline implementation.

Hyperparameter sensitivity: While the authors provided an analysis of parameter impact, the method still appears to exhibit a certain degree of hyperparameter sensitivity.

Outstanding major issue: Crucially, the issues of limited novelty and the lack of discussion with recent related works (e.g., MadaKV, MEDA, SparseMM) remain unsolved. This stands as the major deficiency of the paper.

**Reviewer Scores:**

I think reviewers may maintain their scores.

---

### Decision · Program_Chairs · 2026-01-26

Reject